# Alzheimer’s Disease: From Immune Homeostasis to Neuroinflammatory Condition

**DOI:** 10.3390/ijms232113008

**Published:** 2022-10-27

**Authors:** Lucia Princiotta Cariddi, Marco Mauri, Marco Cosentino, Maurizio Versino, Franca Marino

**Affiliations:** 1PhD Program in Clinical and Experimental Medicine and Medical Humanities, University of Insubria, 21100 Varese, Italy; 2Neurology and Stroke Unit, ASST Sette Laghi Hospital, 21100 Varese, Italy; 3Department of Biotechnology and Life Sciences, University of Insubria, 21100 Varese, Italy; 4Center of Research in Medical Pharmacology, University of Insubria, 21100 Varese, Italy; 5Department of Medicine and Surgery, University of Insubria, 21100 Varese, Italy

**Keywords:** Alzheimer’s Disease, immunity, neuroinflammation, neurodegeneration, homeostasis, microglia, CD4+ T cells

## Abstract

Alzheimer’s Disease is the most common cause in the world of progressive cognitive decline. Although many modifiable and non-modifiable risk factors have been proposed, in recent years, neuroinflammation has been hypothesized to be an important contributing factor of Alzheimer’s Disease pathogenesis. Neuroinflammation can occur through the combined action of the Central Nervous System resident immune cells and adaptive peripheral immune system. In the past years, immunotherapies for neurodegenerative diseases have focused wrongly on targeting protein aggregates Aβ plaques and NFT treatment. The role of both innate and adaptive immune cells has not been fully clarified, but several data suggest that immune system dysregulation plays a key role in neuroinflammation. Recent studies have focused especially on the role of the adaptive immune system and have shown that inflammatory markers are characterized by increased CD4+ Teff cells’ activities and reduced circulating CD4+ Treg cells. In this review, we discuss the key role of both innate and adaptive immune systems in the degeneration and regeneration mechanisms in the pathogenesis of Alzheimer’s Disease, with a focus on how the crosstalk between these two systems is able to sustain brain homeostasis or shift it to a neurodegenerative condition.

## 1. Introduction

Alzheimer’s Disease (AD) is the most common neurodegenerative disease in the world. According to the most recent estimates, approximately 44 million people are affected by dementia, with a prevalence of about 6.4% worldwide, and it is expected that, as the average age of the population rises, this number will increase threefold by 2050 [1,2]. Cognitive impairment leads to serious social and occupational disability [3,4,5]. The global costs related to dementia increased from 604 billion USD in 2010, to 818 billion USD in 2015, thus recording a growth of 35.4%.

AD affects the areas of the Central Nervous System (CNS) involved in thinking, memory, and language with a gradual alteration of all higher cognitive functions, mainly in elderly individuals [6,7]. This neurological condition was first studied and described in 1907 by Alois Alzheimer, when he reported the case of a 51-year-old woman with a progressive cognitive decline, time and space disorientation, and other behavioral changes. Onset is prevalent after 65 years, and it increases with aging, a condition known as LOAD (Late-Onset Alzheimer’s Disease). In 10% of all cases, the early symptoms occur between 30 and 65 years, in relation to a genetic abnormality (EOAD: Early Onset Alzheimer’s Disease). The median life expectancy after the disease is diagnosed is less than 10–12 years.

## 2. Neuropathological Hallmarks of Alzheimer’s Disease

Multiple macro- and microscopic neuropathological features represent pathophysiological hallmarks of AD [8]. The macroscopic distinctive features are defined by symmetrical and diffuse atrophy, with enlargement of both grooves and ventricles and flattening of the cerebral circumvolutions; these changes denote a widespread loss of neuronal mass [9]. At microscopic examination, the two pathognomonic alterations of AD consist of senile plaques (or amyloid plaques) and neurofibrillary tangles (NFT); the first ones are formed by extracellular aggregation of the amyloid-beta peptide (Aβ), which is over-produced from Amyloid Precursor Protein (APP), whereas the second ones are formed by intracellular deposits of hyperphosphorylated tau-protein at the cytoplasmic level.

The presence of neuropathological abnormalities in areas such as the hippocampus and the parietal, frontal, and occipital cortex results in several behavioral and cognitive impairments. The dysfunction of these selected brain areas starts with synaptic damage, which precedes neuronal loss. The reduced number of neurons is consequently associated with neuronal dysfunction [10].

### 2.1. Amyloid Plaques and Amyloid Cascade Hypothesis

One of the main theories for the amyloid plaques’ development is the amyloid cascade hypothesis [11,12,13], which was supported by the evidence of congophilic Aβ-enriched aggregations in the brain tissue of AD subjects [14]. In EOAD patients, these pathological findings could be fully explained by the mutations detected in different genes involved in APP synthesis and metabolism (APP, PS1, and/or PS2 genes) [15,16], and supported by the findings in transgenic mice, expressing at least one of these human mutated genes, that showed the same peculiar cerebral amyloid pathology of AD subjects [17,18]. In recent years, based on these assumptions, many studies have focused both on the role of APP in the brain and the molecular events that promote the amyloidogenic proteolytic cleavage of APP, as depicted in Figure 1 [19].

The amyloidogenic and non-amyloidogenic cleavage of APP is started with the action of β- and γ-secretase, respectively [7,20,21]. In the amyloid-formation pathway, the γ-secretase complex, which is composed by PS1, PS2, and other proteins [22,23], acts on β sub-products-secretase [24,25]. The γ-secretase can operate on three distinct cleavage sites, resulting in Aβ1–38, Aβ1–40, and Aβ1–42, with this latter one showing the higher tendency to aggregate [25]. Increasing evidence suggests that amyloidogenic processing of APP and oligomerization of the Aβ peptides in the CNS are early pathogenic effects that precede and exacerbate tau-associated brain pathology [26,27,28]. Furthermore, the mechanisms involved in the clearance of Aβ from CNS, specifically through the interstitial fluid (ISF) and CSF, are considered to be compromised not only in cases of familial AD, but also in sporadic AD (where there are no obvious differences in brain Aβ-formation rates), enhancing the accumulation of these toxic peptides accumulation in the CNS [29,30,31].

### 2.2. Neurofibrillary Tangles

NFT are constituted by filamentous tau proteins. In AD tau proteins are hyperphosphorylated and abnormally aggregated and lose their usual ability to bind axonal microtubules [32]. This tau’s function loss is matched by an upregulation of abnormal tau aggregation. NFT occur in three different stages and initially appear as “pretangles”, containing abnormal tau (but not polymerized into microscopic clusters) inside neuronal bodies and dendrites. These evolve into aggregated filaments in the soma and proximal cell processes. The mature tangles displace the nucleus and other vital cellular components, and eventually start the neuronal apoptosis. The insoluble filaments are left released in the extra-cellular space, where they associate with astrocytes and microglia. NFT morphology depends on neuron type [33]. They are “flame shaped” in hippocampal pyramidal neurons and the V cortex layer, “globose” in the basal nucleus of Meynert, raphe nuclei, substantia nigra, and locus coeruleus. It has been suggested that their number and location correlate with neuronal loss, disease severity, and clinical course. Additionally, NFTs are more associated with cognitive decline than amyloid deposits alone [34,35].

Aβ plaques and NFT, located in the extracellular space, elicit a pathological cascade, activating microglia and astrocytes, resulting in a neuroinflammatory condition [36]. Microglial cells, through phagocytosis, are able to remove Aβ peptides. Additionally, during AD, the production of proinflammatory cytokines and the release of other damaging substances are highly intensified, contributing to generate a so-called “neuroinflammatory state” [37]. 

This is the main topic of this review.

### 2.3. Alzheimer’s Disease Pathogenesis Theories

The etiology and the pathological factors involved in AD are not completely known, but many studies have suggested that AD pathogenesis is multifactorial and includes genetic, lifestyle, and environmental factors [23,38]

Recent reviews have also considered numerous modifiable and non-modifiable risk factors for AD pathogenesis that seem to act independently from Aβ and tau-pathology [39]. Especially for the LOAD, the proposed different risk factors consist of aging, genetic factors (especially mutation of the APP gene, PSEN1/2 genes, and APOε4 gene), exposure to aluminum, head injury, diet, smoking, mitochondrial dysfunction, vascular disease, epileptic activity, immune system dysfunction, and infectious disease may result in the impairment of cognitive function due to neurotransmitter disruption [38,40]. In the recent years, many studies suggested that the hippocampus can be injured by long-term microwave exposure, and many arguments relay the possibility that microwaves may be involved in the pathophysiology of CNS disease, including AD [41,42,43].

Past studies proposed three main competing hypotheses to explain the pathogenesis of AD:

-First, cholinergic theory, which suggests that AD is caused by a degenerative process that is capable of selectively damaging groups of cholinergic neurons in the hippocampus, frontal cortex, amygdala, nucleus basalis, and medial septum, regions and structures that serve important functional roles in attention, learning, and memory. This selective alteration leads to the reduction of cholinergic markers such as acetylcholinesterase [44]-Second, amyloidogenic theory, which suggests that an abnormal clearance of amyloid-beta protein induces the accumulation of amyloid β in the cerebral neurons, leading to a neuronal impairment and to increased neuronal apoptosis [12,45];-Third, tauogenic theory, which proposes that tau protein aggregation and, consequently, NFT development, directly cause neuronal abnormalities, activating a neuroinflammatory condition in the extracellular space and inducing neuronal apoptosis [46].

Considering the amyloidogenic theory more plausible compared to the others, in the past years, the research for a possible treatment for AD focused on reducing Aβ plaques and the enzymes involved in amyloid processing, using disease-modifying therapies targeting amyloid (such as β-secretase inhibitors, γ-secretase modulators, Bapineuzumab, etc.) [47,48,49]; however, the results showed that treatments were not curative and unable to affect AD clinical course or the underlying disease’s neuropathology [50]. It is possible that the failure of these treatments can be partially attributed to the so-called “Lesion Seduction Concept”, i.e., a simplistic paradigm, which assumes that the AD histopathological lesions are a direct reflection of its etiology [51].

Currently, however, there is increasing evidence of an early involvement of other pathological mechanisms which begin long before the formation of amyloid Aβ- and tau-protein hyperphosphorylation [52]. These mechanisms include a chronic immune-mediated neuroinflammation state and a pathological cerebral aging, called “neuroinflammatory-state” [37]. Recent studies showed that these mechanisms, linked with the innate and adaptive immune system activities, are likely to play a major role in the pathogenesis of different neurodegenerative diseases, including AD.

## 3. The Role of CNS Immune System

The concept of “CNS as immune privileged site” [53,54] was founded based on its limited abilities to resist to injury, during inflammation, and its poor capacity to regenerate (regenerative ability) [55,56]; numerous evidences on the presence of afferent and efferent connections, between the CNS and the peripheral immune system are available [57,58,59].

The blood–brain barrier (BBB) bound brain tissue and checks the peripheral immune cells’ entry [60]. The monitoring of immune cells infiltration into CNS is mediated by cell-adhesion molecules (CAMs) and CAM ligands on BBB endothelial cells. During neuroinflammatory responses, CAMs expression can be upregulated, and this condition results in the ability of adaptive immune cells, mainly CD4 + T cells, to cross the BBB [61,62] and interact with the brain’s resident immune cells, such as microglial and astrocytes [63,64,65,66,67,68,69,70].

Another issue concerns the neuronal-waste management through the so-called “glymphathic” system [71,72,73]. Cerebral Spinal Fluid (CSF) is able to pass from the subarachnoid space through the arterial perivascular space to the brain interstitium, draining neuronal cellular waste, through the aquaporin water channels [74]. The CSF flow goes toward venous perivascular space and takes out neuronal waste into meningeal lymphatic vessels and then drains into lymphoid tissue, especially in cervical lymph nodes [75]. The glymphatic system can promote CNS clearance of lipophilic and hydrophilic substances and plays a critical role in removing neurotoxic protein aggregates, such as (Aβ) plaques. Therefore, dysfunction in this system would lead to an Aβ accumulation, and to AD [30]. This is in line with the findings by Van Zwam et al. who have shown that the surgical removal of CNS draining lymphnodes (deep cervical lymph nodes) significantly exac-erbates the severity of the neurodegenerative diseases [76].

### 3.1. Physiological Role of CNS Innate Immune Cells

As mentioned before, microglia constitute more than 80% of resident immune cells. In normal conditions, these cells play a key role in cerebral circuit development and synaptic homeodynamics [77,78].

Microglial cells have a crucial role in presynaptic microenvironment immune surveillance and also in synaptic remodeling, leading to axonal and dendritic terminal pruning, by modulating proteolytic and phagocytic processes. Microglial cells recruit astroglia, or they can be recruited by astroglia [79]. They express a large array of receptors that detect exogenous or endogenous CNS insults and are able to start an immune response. In addition to their typical role, as immune cells, microglia protect the cerebral tissue by providing phosphocyte clearance and trophic sustenance to support brain repair. Therefore, microglia have a crucial role in brain tissue because they are involved in monitoring and preserving the homeostatic environment and are able to defend and remodel synapses in order to maintain the essential plasticity of neuronal pathways [80]. This effect is enhanced by the release of trophic factors, including brain-derived neurotrophic factor (BDNF), which is also involved in memory pathways [81]. Once primed by pathological triggers, such as neuronal death or protein aggregates, microglia become activated and then begin to migrate to the lesion site and start an immune response.

When microglia is activated by these “warning signals” from cerebral tissue, it enters the so-called “activated microglia state” [82], which involves the M1 microglia phenotype [83,84,85] that consists in morphological cytoskeleton changes in modification of the molecular mediators’ releasing profile and in increased proliferative responses [86]. This activates state sets a “frontline of the fight” against alterations in the brain’s homeostasis by providing an interplay between cytotoxic or neuroprotective factors [83].

The M1 or “pro-inflammatory or “activated” microglial phenotype is able to release proinflammatory cytokines such TNF-α, IL-1β, IL-12, and also nitric oxide. It is able to reduce the release of neurotrophic factors, thus enhancing inflammation and cytotoxicity. On the other hand, the “M2 or anti-inflammatory” phenotype secretes anti-inflammatory cytokines, such IL-10, IL-13, and IL-4, raising the expression of neurotrophic factors (brain derived neurotrophic factor-BDNF, TGF-β) [83] and multiple signals implicated in the protection and reparative processes and downregulation of inflammatory responses [36].

### 3.2. Role of CNS Innate Immune System in Alzheimer’s Disease

As mentioned above, it is well-known that the immune system contributes to maintaining the CNS homeostasis and CNS-innate immune resident cells, macrophages and all microglial cells, are an important active components of brain aging, neuroinflammation and different neurodegenerative diseases [87], either through cytokines production and phagocytic activity or as a result of adaptive immune system stimulation [88,89].

In recent years, several studies have confirmed that microglia activation is one of the key components, related to the progression of the principal neurodegenerative disorders (for instance Parkinson’s Disease PD, AD and fronto-temporal dementia-FTD) [81,83,85,90].

As previously mentioned, microglia is responsible for monitoring and keeping the homeostatic environment [91]. At the same time, microglia contributes to the synapses’ protection and remodeling aimed at preserving a neuronal circuit plasticity [80], also through the Brain Derived Neurotrophic Factor action, which is involved in memory process [81].

Interestingly, some studies have shown that the microenvironment where microglia interact with neurons is different between the different neurological disorders and between different CNS regions [92], suggesting that different neurological conditions occur in different brain areas, which, in turn, have different microglial cells activity. Batchelor et al. demonstrated that, after a mechanical injury in the CNS, there is a greater inflammatory response in the spinal cord compared to brain and an increased inflammatory reaction in white matter as compared to grey matter [93].

#### 3.2.1. Microglia and Aging in Alzheimer’s Disease

Many studies showed the effect of aging in microglial cell response [94]. Aging is considered the main risk factor for several neurodegenerative disorders, including AD [95,96], and it has been demonstrated that it activates some changes modified in gene expression in microglial cells, resulting in an aberrant cytoplasmatic formation and fragmented processes and possibly influencing the disease’s development. In aged mouse models, microglia have been found to have reduced expression of β -amyloid degrading enzymes and reduced phagocytosis [97] and AD microglial function is impaired by the presence of β -amyloid aggregates, thus leading to a self-perpetuating cycle of increased b-amyloid accumulation and further damage [98]. 

#### 3.2.2. Microglia and Aβ-Protein in Alzheimer’s Disease

Many recent studies demonstrated that different microglial profiles and phenotypes are associated with different neurodegenerative diseases and their different phases of progression. 

In AD experimental models, it has been shown that microglia cells, which surround amyloid plaques, through a chemotactic process, are able to eliminate the plaques, reduce their growth and accumulation in extracellular spaces [86,99]

However, in AD, a persistent homeostatic alteration, such as the accumulation of Aβ in CNS, can be an activator trigger called “priming” [100]. Priming activates microglia, making them apt for inflammatory stimulating factors, which can then result in amplified inflammatory reactions [101]. The sustained exposure to Aβ itself, chemokines, cytokines, and other inflammatory mediators seems responsible for the persistent functional impairment of microglial cells, as observed at plaque sites [86]. Activated microglia represent a typical pathophysiological hallmark of AD [101,102].

Additionally, the disregulated microglia activity (also known as dystrophic microglia) could be either a priming factor or a worsening factor or both, of abnormal Aβ deposition in CNS [36]. Recent studies have shown how microglia might contribute to the accumulation of Aβ plaques [103,104]. Initially in AD, microglia might be able to phagocytize soluble amyloid-β plaques and then accumulate them in intracellular space. In past studies it has been shown that Aβ protein can build-up within microglia, gaining resistance to elimination and degradation by microglia itself [105]. In the presence of proteins, associated with cell apoptosis (i.e., speck-like protein), microglia becomes ineffective in destroying Aβ plaques and undergoes cell death [104]. This condition support the hypothesis that microglia apoptosis may have a role in increasing plaque formation [90]. The Aβ aggregates themselves induce a process of neuroinflammation and neurodegeneration by stimulating microglia to produce and release cytokines and also by interfering with the production of anti-inflammatory cytokines (Figure 2b) [106,107]. Moreover, these protein aggregates are able to promote neuronal dysfunction and apoptosis by activating inflammation and oxidation, mediated by CNS microglia and astrocytes [108,109].

### 3.3. Physiological Role of Peripheral Adaptive Immune Cells 

The adaptive immune system consists of two principal cellular effectors of immune responses: the first is represented by T lymphocytes, which develop in the thymus, and the second by the antibody-producing cells, called B lymphocytes, which originate in the bone marrow [110].

After growing in primary lymphoid organs (bone marrow and thymus), lymphocytes move to the secondary lymphoid organs (spleen and the lymph-nodes), where the adaptive immune responses start and are modulated by innate immune signals.

T cells play an important role in pathogens’ elimination since they can directly, through the direct cytotoxic action of CD8+ T cells, or indirectly kill infected target cells through the action of the most represented type of T cells: CD4+ cells. Most of these are labeled as Th-cells because they show a “helper function”: they have no phagocytic activity, and they are unable to kill pathogens directly; they indirectly do so through the activation of other cells (Natural Killer/CD8 T cells). CD4+ T cells can also be triggered by peptides presented by MHC-II complex on macrophages, B cells, and dendritic cells [82]

CD4+ T-cell receptors (TCRs) bind to peptides complexed with class II MHCs (HLA-DQ, HLA-DP, and HLA-DR). Class II MHC molecules are found on APCs and are induced by innate immune stimuli, such as TCR ligands. The recognition of peptide-MHC APC complex by TCR-induced T-cell activation leads to a rapid aggregation of TCR-associated molecules between T cells’ surfaces and APCs and a consequent condition called “immunologic synapse” [111].

There are four principal categories of Th cells: Th1, Th2, Th17, and T-regulatory (Treg) cells. Th1 lymphocytes release pro-inflammatory cytokines (IL-2, IFN–γ, and TNF-α) that activate macrophages and cytotoxic T cells’ CD8+ in order to destroy intracellular bacteria and virus-infected targets; Th2 cells produce anti-inflammatory cytokines (IL-4, IL-5, and IL-13) that activate B cells and elicit antibodies’ production, as well as hypersensitivity and parasite-induced immune responses; and Th17 cells release pro-inflammatory cytokines (IL-17 and IL-22) and cytokines/chemokines that promote the activation of neutrophils and macrophages.

Th1 cells are identified by the ability to differentiate from their naive Th0 precursors, due to IL-12 and IFN-γ secretion and the influence of T-box transcription factor (T-bet) expressed in T cells. Conversely, Th2 cells release IL-4, IL-5, IL-10, and IL-13, and their growth is mediated by IL-4 and the transcription factor GATA-3. IL-6, TGF-β, and the expression of transcription factor ROR-γt (retinoic acid receptor–related or-phan receptor γt) promote the activation of Th17 lymphocytes [111,112,113,114]. Th17 cells produce IL-17, a group of five homologous molecules designated as IL-17A-F. Th17 cells release, in turn, IL-17A and IL-17F cytokines. These latter are powerful pro-inflammatory citokynes that are able to produce IL-6 and TNF, as well as leading granulocyte recruitment. Th17 cells play an important role in autoimmune disorders and in inflammatory allergic processes, such asthma [112]. IL-17 is present in the in-flamed tissues of patients with arthritis, multiple sclerosis, and systemic lupus erythematosus. In animal models, genetic deletion or antibody inhibition of IL-17 blocks experimental autoimmune diseases, such as experimental autoimmune encephalomyelitis. Conditions that result in the circulating Th17 reduction are associated with a poor inflammatory response and the development of recurrent infections [112].

Treg cells are essentially immunosuppressive, anti-inflammatory.

The critical activity of Treg-cell responses is also present inside the CD4+ abTCR subgroup of T cells and is probably performed by different types of regulatory cells. Both IL-10-producing Treg cells and CD25+ CD4+ T cells expressing the forkhead box protein 3 transcription factor (FOXP3) are able to reduce T cells’ responses. Treg lymphocytes exert an anti-inflammatory activity and promote immune cells’ suppression in order to maintain immune homeostasis. It has been shown that the absence of FOXP3, encoded on the X-chromosome, can cause a severe multisystem inflammatory deficit (referred to as immune dysregulation, X-linked syndrome) [115,116]. In addition to CD4+ T cells, CD8+ T cells are an important subset of circulating T cells and are able to identify peptide antigens, which are presented by the MHC-I complex and then re-lease cytotoxic granules, against the recognized cells [97,117].

### 3.4. Role of Peripheral Adaptive Immune System in Alzheimer’s Disease

Microglia is not completely responsible for all AD pathology, and the role of peripheral adaptive immunity has been acknowledged [63,118,119].

The interplay between the innate and adaptive immune system is crucial for the relationship between neuroinflammation, neurodegenerative and neuroprotection [63]. In this regard several studies suggest that microglial cells are responsible for interacting with the adaptive peripheral immune system: cell-mediated immunity, performed by both pro-inflammatory (Th1 and Th 17) and anti-inflammatory (Th2 and Treg) T cell subtypes, could further regulate the activity of microglia themselves, promoting a neurodegenerative -M1- or neuroprotective -M2- phenotype [63,120].

In CNS peripheral immunity cells, in particular T cells, B cells, dendritic and natural Killer (NK) cells, located in the brain tissue, in the meninges and in choroid plexus, are responsible for either neuroprotection or disease instigation, according to the local environment [121,122]

T lymphocytes are cells, developed in the thymus, through the expression of TCR and glycoproteins, that respond to immune stimulations, as a part of cell-mediated and humoral immunity of adaptive immune system [123,124,125]. 

A correct balance between anti and pro-inflammatory activities, is required for maintaining CNS homeostasis and brain healthy (Figure 2a). 

In pathological conditions, T cells move on in peripheral blood and switch to an activated state, triggered by the recognition in lymphoid organs of their TCRs, in the context of MHC molecules; or by antigen-presenting cells (APCs). T cell activation results in a massive proliferation and clonal expansion of T cells, triggered by the following effector functions: cell-cell contact, cytokines production, B cells activation, cell death induction or innate immune cells modulation [69,126]. Astrocytes and microglia are able to govern T cells activation (“Priming T cells”) and differentiation by cytokines or molecules, including IL-1, IL-6, TNF-α, IL-10 and TGF-β, responsible for Treg and Teff differentiation [127,128]. Teff influences and maintains a pro-inflammatory microglia phenotypes, via secretion of IFN-γ and IL-17 or release of granzyme B [129,130]. Treg activates neuroprotective responses. Treg has an important function in antigen-specific immune tolerance, deleting effector responses against a different range of antigens, including antigens from self, from bacteria and from the environment [131,132]. Treg also reduces Teff function and proliferation through several mechanism: by releasing immunosuppressive cytokines (TGF-β, IL-10, IL-35) or granzyme B and perforin 1, that induce apoptosis and cytotoxicity. Treg is also able to promote “regenerative activities” especially in tissues, that include kidney, retina and skin. This was demonstrated by several studies, that showed an increase of tissues damage and a reduction of vascular repair, after a Treg depletion [133,134,135,136].

In the brain environment if CD4+ and CD8+ T cells fails to detect they get back to the periphery via CNS lymphatic system, into deep cervical lymph nodes. Otherwise they start a local effector immune-reaction [85,137,138]. During neuroinflammatory conditions, the interaction between immunity cells and related-antigens results in an alteration of cell trafficking and secretory ability of the BBB [139]. BBB damage bring to an altered transporter and cytokines responses, that leads to an excessive migration of immune cells into the brain tissue with an amplification of inflammatory condition [89,122,140,141]. Immune responses upregulate molecules trafficking, cell transmigration and neuronal good function, when antigen-specific CD4+ T cells cross the BBB [142]. When T lymphocytes do not detect specific antigen, they do not penetrate the BBB and consequently undergo to apoptosis [143]. T cells are able to l regulate brain homeostasis, through a cascade of immune signals and secretory molecules, even without crossing the BBB [144].

In a healthy brain all CD4+ T cells subtypes contribute to guarantee an homeostasis, although, during neurodegenerative diseases such as AD, Th1, Th2, Th17 and Treg play different roles in neuroprotection and neurodestruction [51). During aging phase, several studies have demonstrated that there is an increase of all lymphocytic subtypes in the brain tissues [145,146]. The role of T cell autoimmunity has been studied in different animal models [147,148,149,150], but less commonly in human brain tissues [151,152]. In AD animal models with mutations of APP and consequently elevated levels of Aβ, an infiltration by an increased number of overall T cells in the CNS has been observed, along with a concomitant upregulation of endothelial adhesion molecules ICAM-1 and VCAM-1 [153,154,155,156]. Also in post-mortem AD patients brain tissues, many studies have shown increased numbers of CD4+ and CD8+ T lymphocytes of AD patients [157], compared to healthy controls [158,159,160,161,162]. There are, however, many evidences to hypothesize that T cell activities in neurodegenerative disorders, especially in AD, are directed against aggregated or misfolded proteins [119]. During the initial AD stages the altered BBB promotes the entrance of T cells, including Treg, that contribute to CNS immunity protection [132,163]. In advanced disease stages the immunosuppressing activities of Treg are reduced, by the production of IFN-γ, released by Th1, and IL-17, released by Th17 cells, which therefore affect a breakdown of immune-tolerance [164,165]

In recent years, different studies considered the mechanism of altered regulation of CD4+ T-cell activity in neurodegenerative diseases. For instance, in PD. Kustrimovic and coworkers [166] showed reduction of circulating CD4+ T cells, especially Th2, Th17 and Treg with a shift of CD4+ T cells towards the Th1 lineage. In comparison, however, the role of peripheral adaptive immunity in AD pathogenenesis has received less attention especially regarding the pattern profile of immune dysregulation. Instead few studies had demonstrated that AD patients’ peripheral immune profile showed significant aberrations in immune cells, which may be associated with the progression and different phases of AD [162,167,168].

The studies on this topic in AD (both in human and in animal models are summarized in Table 1).

Saresella et al. showed a elevated levels of RORy Th17 cells in peripheral blood of AD patients, compared to those with Mild Cognitive Impairment (MCI) and age-matched healthy subjects [168]. The same Th17 increase was reported by Agnes Pirker-Kees, 2013 [169] and by Ciccocioppo F et al [170] and Gate D et al [97]. Heneka et al showed that IL-17, the key cytokine produced by Th17, was able to weaken BBB tight junctions and to promote peripheral leucocytes entry in to the brain tissue, inducing an inflammatory response, mediated by IL-1β, IL-6 and TNF-α [171].

In mouse models the injection of Aβ into the hippocampus causes Th17 infiltration and an upregulation of IL-17/IL-22 in the hippocampal tissue, in the CSF and in the blood [171,172,173]. Th17 was thought to promote neurodegeneration through direct activation of Fas-FasL apoptosis [172].

The data about circulating Treg are more controversial, since their level was found to be reduced [97,170] or normal [174]. Tregs have an inflammatory function and their neuroprotective role has been largely demonstrated: Treg cells are able to delay the progression of AD, ad Treg reduction worsens cognitive decline [175]. In APP/PS1 mice the increased number of circulating Treg lymphocytes decreased cognitive decline and increases microglia in plaques [176]; Treg depletioninduces a decline of both memory and microglial activities [176]. Tiemessen et al. showed that Treg also induces the microglia conversion from M1 to M2 phenotype, an effective mechanism to reduce neuroinflammation [177]. Studies on AD mouse models have shown that CD4+CD25+ Treg lymphocytes are the main modulators of immune responses, maintaining an immunological tolerance to self antigens, slowing down the progression of AD and modulating the microglial response to amyloid deposition [178]. Treg immunosuppression affects CNS innate immune cells phagocytic activities and consequently amyloid plaques clearance in mouse model [179].

Baruch K et al. have demonstrated that the blood peripheral reduction of circulating Treg cells is consequently followed by their increased in CNS tissues, with an enhanced anti-inflammatory activity, which could suggest that peripheral and tissue-infiltrating Tregs play distinct roles in CNS disorders. Although a complete suppressing would be harmful, a transient depletion of peripheral Foxp3-Treg cells or pharmacological inhibition of their activity, can be able to mitigate central neuroinflammatory response and to improve cognitive decline, by increasing the clearance of Aβ and thus reducing plaque formation [179]. Depletion of circulating Treg lymphocytes has been shown to reduce recruitment of activated microglia to amyloid deposits, without changing β-amyloid clearance by microglia themselves [176]. These last evidences suggest that a reduction of circulating Treg may have a neuroprotective role and help to delay the progression of AD pathology by minimizing the reduce of other T cell subtypes [180] (Table 1).

**Table 1 ijms-23-13008-t001:** Summary of peripheral-immune-cell-profile evaluation studies in Alzheimer’s Disease condition (in both animal and human models).

Tissue	Species	Increased CD4+	Increased Th17	Reduced Treg	Increased CD8+	Ref
AD Postmortem brain tissue	human	yes	no	no	yes	[149]
AD Postmortem brain tissue	human	yes	no	no	yes	[151]
AD Postmortem brain tissue	human	yes	no	no	yes	[152]
AD Postmortem brain tissue	human	yes	no	no	yes	[150]
AD Postmortem brain tissue	human	yes	no	no	no	[136]
Brain tissue and peripheral blood	transgenic APP mouse	yes	no	no	no	[145]
Brain tissue/peripheral blood	transgenic APP rat/human	yes	no	no	no	[146]
Peripheral blood	human	no	no	yes	no	[155]
Brain tissue	transgenic APP mouse	yes	no	no	no	[144]
Peripheral blood	human	no	yes	no	no	[160]
Brain tissue	Mouse	no	yes	no	no	[163]
Brain tissue	5xFAD AD mouse	no	no	no	no	[170]
AD Postmortem brain tissue	human	yes	no	no	yes	[148]
Brain tissue	transgenic APP mouse	yes	no	no	no	[147]
Brain tissue	3XTg AD mouse	no	no	yes	no	[166]
Brain tissue/peripheral blood	transgenic APP1 mouse	no	no	yes	no	[167]
Brain tissue	mammalian	yes	no	no	no	[137]
Peripheral blood	human	no	no	no	no	[165]
Peripheral blood	human	no	yes	yes	yes	[161]
Peripheral blood	human	no	yes	yes	yes	[90]

In conclusion, these data can support the notion that Tregs play a crucial role in the pathogenesis of AD; more specifically, the Treg activity seems to have a “neuroprotective role”, both directly with related cytokines and indirectly with its effects on innate immune cell functions. These studies also suggest that the analysis of peripheral blood immune profile is a candidate to represent an additional biomarker of neurodegenerative disorder, especially for AD, that could be used in the future to better characterize the early diagnosis of AD [139].

## 4. Discussion

This review shows that innate and adaptive immune systems contribute to the inter-play between neuroinflammation, neuroprotection, and neurodegeneration mechanisms involved in AD pathogenesis.

In the healthy brain, there is a perfect homeostatic condition between anti- and pro-inflammatory mediators and also between peripheral and innate immune systems. In contrast, in Alzheimer’s Disease, as well as in other neurodegenerative, vascular, infectious, or metabolic disorders [65,66,67], innate and adaptive immune systems are often dysfunctional, with altered peripheral levels of immune cells and an unbalance that favors pro-inflammatory components and, thus, neurodegeneration [115,116]. In particular, in AD, the brain homeostasis switches to an unbalanced condition, characterized by an “activated microglia” state triggered by the amyloid plaques and NFT. These AD hallmarks are able to directly activate the pro-inflammatory response of adaptive peripheral immune cells through APCs in the peripheral tissues. T cells are able to regulate brain homeostasis through a cascade of immune signals and secretory molecules, even without crossing the BBB. Thus, AD pathogenesis has been proven to be related to the emergence of effector immune populations and expanded inflammatory activities [67,169]. This evidence underlines the complexity of the brain’s microenvironment, which fluctuates between inflammatory and anti-inflammatory states on a continuous cycle.

Many recent studies have also led to the knowledge that there is a component of the adaptive immune system that is represented by Treg cells and is also involved in neurorepair and regenerative activity [54]. Treg cells have an important function in antigen-specific immune tolerance, deleting effector responses against a different range of antigens, including antigens from the self, and promoting “regenerative activities” in tissues [133]. In AD, many studies have shown that inflammatory markers are characterized by increased CD4+ Teff cells’ activities and reduced circulating CD4+ Treg cells [164,181]. Therefore, the outbreak of peripheral effector immune cells and downregulation of regulatory immune cells could represent an early peripheral-blood-disease biomarker. The switch between degenerative and regenerative conditions reveals a need to expand this field of research to improve future therapeutic approaches.

The awareness of immuno-surveillance in the CNS has paved the way for the identification and characterization of peculiar inflammatory responses in many CNS diseases—especially in AD—that were previously considered to be exclusively degenerative diseases.

## 5. Conclusions

The key role of innate and adaptive immune system in the pathogenesis of AD suggests the importance of discovering new therapies and treatments that can modify the clinical course, especially in the preclinical stages, where the brain is still preserved. In the past, immunotherapies for neurodegenerative diseases have focused on Aβ plaques and NFT treatment. Instead, in the few last years, the therapeutic strategies have aimed to promote expansions of immunotherapies, focused on immunoregulatory, neuroprotective, neurodegenerative, and anti-inflammatory Treg activities.

In the future, this approach will constitute a promising field of research, with the aim of “fighting” the neurodegenerative disorders’ progression and supporting the neuroprotective role of the immune system.

## Figures and Tables

**Figure 1 ijms-23-13008-f001:**
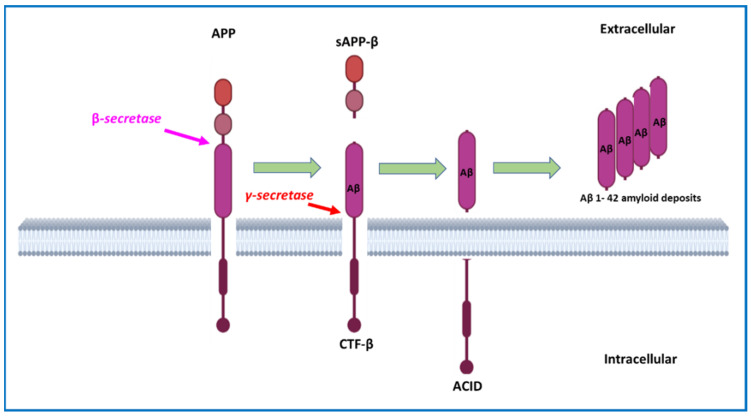
The process of Aβ1–42 formation and plaque deposition. Schematic diagram of the progressive cleavages of the amyloid beta (Aβ) precursor protein (APP) transmembrane domain. Aβ peptide is generated from APP processing via the amyloidogenic pathway, by the β and γ-secretases complex, which produces a peptide called sAPP-β (soluble ectodomain of APP-produced by β secretase) and CTF- β (C-Terminal Fragment by β secretase) fragment and foremost ACID peptide and different lengths of Aβ peptides, including Aβ42, which is more prone to aggregation and plaque formation than Aβ40 and has stronger neurotoxicity and (see the text below for more details).

**Figure 2 ijms-23-13008-f002:**
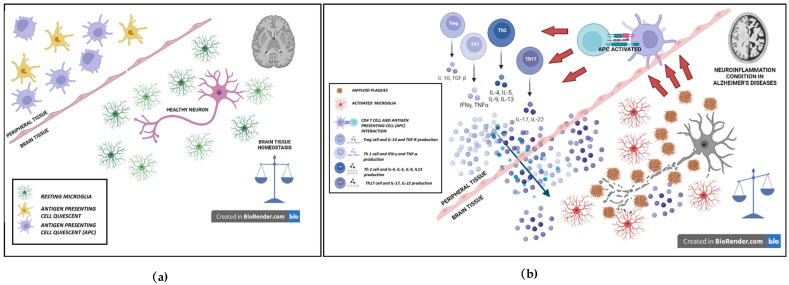
Role of innate and adaptive immune system in Alzheimer’s disease: (**a**) The role of innate and adaptive immune systems in CNS homeostasis. In healthy brain tissue, there is a condition characterized by homeostasis and balance between normal neuronal cells; innate immune resident cells, in particular, quiescent microglial and astrocytic cells, which are called “resting microglia” and “resting astrocytes”; and immune peripheral tissue cells, such as quiescent antigen-presenting cells (APCs) that are not activated. In this phase the cerebral environment does not present an inflammation-related activation. (**b**) The role of innate and adaptive immune systems in Alzheimer’s Disease neuroinflammation condition. Upon several different types of insults, such as genetic, lifestyle, medical, environmental, or psychiatric disorders, healthy neurons become damaged, releasing amyloid plaques and *NFTs* or self-antigens. These antigens remain in the CNS and stimulate resting microglia, making them an activated phenotype microglia that produce pro-inflammatory mediators such as cytokines, reactive oxygen, and nitrogen species, thus increasing oxidative stress and further increasing the neuronal damage. Misfolded self-proteins are processed and then presented on MHC by APCs to naive to cells in lymph-node tissues. Upon recognition of antigens, T cells differentiate into antigen-specific T-effector (Teff) or T-regulatory (Treg) cells. Teff subsets include Th1, Th2, and Th17. Upon identification of modified self-antigen, activated Teff cells generate neurotoxic and proinflammatory cytokines that drive resting microglia to a reactive state and support a neurotoxic cascade. Th1 and Th17 T cells produce neurotoxic cytokines, such as TNF-α, IL-17, IL-22, and IFN-γ, which are released into brain tissue, promoting an enhanced inflammatory cascade. In response to inflammatory events, Tregs attempt to balance neurotoxic activation through the inhibition of antigen presentation and the production of anti-inflammatory cytokines such Il-10 and TGF-β, with the aim to stop the neuroinflammatory/neurodegenerative condition and promote neuronal homeostasis.

## Data Availability

Not applicable.

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
