# Peer review of "Alzheimer’s Disease: From Immune Homeostasis to Neuroinflammatory Condition"

_ijms, 2022, doi:10.3390/ijms232113008_

Round 1

Reviewer 1 Report (Previous Reviewer 2)

The study by Lucia et al. reviewed role of both innate and adaptive immune systems in neuroinflammatory pathways of Alzheimer’s Disease progression. Firstly, this manuscript does not provide crucial update in the field in additional to existing literature (see below). However, it could be an additional article emphasizing systemic nature of AD pathology.

Heneka, M. T., Carson, M. J., El Khoury, J., Landreth, G. E., Brosseron, F., Feinstein, D. L., ... & Kummer, M. P. (2015). Neuroinflammation in Alzheimer's disease. The Lancet Neurology14(4), 388-405.

Ardura-Fabregat, A., Boddeke, E. W. G. M., Boza-Serrano, A., Brioschi, S., Castro-Gomez, S., Ceyzériat, K., ... & Yang, Y. (2017). Targeting neuroinflammation to treat Alzheimer’s disease. CNS drugs31(12), 1057-1082.

Bronzuoli, M. R., Iacomino, A., Steardo, L., & Scuderi, C. (2016). Targeting neuroinflammation in Alzheimer’s disease. Journal of inflammation research9, 199.

Hensley, K. (2010). Neuroinflammation in Alzheimer's disease: mechanisms, pathologic consequences, and potential for therapeutic manipulation. Journal of Alzheimer's disease21(1), 1-14.

Onyango, I. G., Jauregui, G. V., ÄŒarná, M., Bennett Jr, J. P., & Stokin, G. B. (2021). Neuroinflammation in Alzheimer’s disease. Biomedicines9(5), 524.

Comments:

While the content in the review is satisfactory but the manuscript is filled with typo error, spelling error, vague sentences, colloquial sentences, inconsistencies in abbreviation, grammatical error that needs to be corrected.

Further, Authors have not implemented any changes suggested in the 1st round of review

Figure 1 is rudimentary ( doest not provide any new information or don’t show the complexity of the abeta formation) . It should be redrawn with more details.  

Figure 2 resolution need to be improved. In Fig 2 a,b depict in a way to show the change between healthy and damaged neuron. Currently neuron is healthy in both 2 a and b. Increase the resolution of the figures so that the cytokine names are visible. Also give the figure shape legend details for microglia types and other cell types illustrated.

Rigorous proofreading of the manuscript by authors is “must” for further review.  

To mention few (there are several errors in your manuscript):

Thanks to, generally speaking – colloquial sentences

In line 35, it is 35.4% not 35,4%

In line 223, T-cell receptor (TCR) not Toll- like receptors. Wrong abbreviation is mentioned

Be consistent with naming (sometimes Alzheimer’s disease sometimes Alzheimer Disease,

sometimes T reg sometimes Treg, sometimes PS sometimes PSEN, sometimes EOAD sometimes EOD, sometimes microglial sometimes microglia)

Author Response

The study by Lucia et al. reviewed role of both innate and adaptive immune systems in neuroinflammatory pathways of Alzheimer’s Disease progression. Firstly, this manuscript does not provide crucial update in the field in additional to existing literature (see below). However, it could be an additional article emphasizing systemic nature of AD pathology. 

Heneka, M. T., Carson, M. J., El Khoury, J., Landreth, G. E., Brosseron, F., Feinstein, D. L., ... & Kummer, M. P. (2015). Neuroinflammation in Alzheimer's disease. The Lancet Neurology14(4), 388-405.

Ardura-Fabregat, A., Boddeke, E. W. G. M., Boza-Serrano, A., Brioschi, S., Castro-Gomez, S., Ceyzériat, K., ... & Yang, Y. (2017). Targeting neuroinflammation to treat Alzheimer’s disease. CNS drugs31(12), 1057-1082.

Bronzuoli, M. R., Iacomino, A., Steardo, L., & Scuderi, C. (2016). Targeting neuroinflammation in Alzheimer’s disease. Journal of inflammation research9, 199.

Hensley, K. (2010). Neuroinflammation in Alzheimer's disease: mechanisms, pathologic consequences, and potential for therapeutic manipulation. Journal of Alzheimer's disease21(1), 1-14.

Onyango, I. G., Jauregui, G. V., ÄŒarná, M., Bennett Jr, J. P., & Stokin, G. B. (2021). Neuroinflammation in Alzheimer’s disease. Biomedicines9(5), 524.

Comments:

While the content in the review is satisfactory but the manuscript is filled with typo error, spelling error, vague sentences, colloquial sentences, inconsistencies in abbreviation, grammatical error that needs to be corrected.

Further, Authors have not implemented any changes suggested in the 1st round of review For each revision we made all the changes required by the reviewers, and we sent a letter to show how we handled each point by point and how we modified the text accordingly.

Figure 1 is rudimentary ( doest not provide any new information or don’t show the complexity of the abeta formation) . It should be redrawn with more details.   the figure 1 was revised according to the  reviewer comments.

Figure 2 resolution need to be improved. In Fig 2 a,b depict in a way to show the change between healthy and damaged neuron. Currently neuron is healthy in both 2 a and b. Increase the resolution of the figures so that the cytokine names are visible. Also give the figure shape legend details for microglia types and other cell types illustrated.

 The figure 2 has been completely modified

Rigorous proofreading of the manuscript by authors is “must” for further review.  

To mention few (there are several errors in your manuscript):

Thanks to, generally speaking – colloquial sentences . The sentence was corrected

In line 35, it is 35.4% not 35,4% Corrected

In line 223, T-cell receptor (TCR) not Toll- like receptors. Wrong abbreviation is mentioned Corrected

Be consistent with naming (sometimes Alzheimer’s disease sometimes Alzheimer Disease,

sometimes T reg sometimes Treg, sometimes PS sometimes PSEN, sometimes EOAD sometimes EOD, sometimes microglial sometimes microglia)  The sentences were corrected

Moreover the entire test underwent English revision,

Reviewer 2 Report (New Reviewer)

This manuscript asks an interesting, but ambitious scientific question that is worth addressing: Are “dysfunctions” in Down Syndrome caused by a global systemic differential disruption of gene expression across the genome; or by gene dosage?  Although it is unlikely that their set of experiments can answer the question at hand, this study does provide a valuable set of empirical data.  That is, the gene expression would have low correlations when the brains are in a developmental stage at 0-12 months are compared with those that are in matured or aging stages at 30-39 or 40-42.  This study provides valuable insight into those who are working on stem cell work for its implications. 

Comments.

·       This manuscript needs editing by a native English speaker.  For example, the first sentence in the Background in the Abstract is an incomplete sentence.

·       This is an ambitious paper to answer what the main mechanism of Down syndrome is.  However, it is doubtful that a set of transcriptomic data can answer that.  The manuscript would be more informative if the authors tighten the questions. 

·       Table 2 doesn’t make sense in that the ‘brain’ has 486 genes that are differentially expressed; yet, the number of genes that are differentially expressed in different sections of the brain range from 415 to 601.  This is logically impossible, since all brain sections are part of the brain, the number of gens that are differentially expressed in the brain must be 601 or greater. While ‘the Brain’ has an asterisk, there is no text that goes with the asterisk to explain how this term was defined.  

·       It is not at all surprising that ‘the brain’ will not have high correlation with some of the sections, since ‘the brain will be weighted average over all sections in the brain which will be quite heterogeneous. 

Minor points.

·       Page 6. Para 2, line 227.  Inferior temporal cortex should be ITC, not Cerebellar Cortex.  Since cerebellar cortex is mentioned in line 223, this must be an error.

Author Response

This is not a review report form of the manuscript ijms-1865271 with the title of 

"Alzheimer’s Disease: from Immune Homeostasis to Neuroinflammatory condition. A Review".

Reviewer 3 Report (New Reviewer)

Review on Alzheimer’s Disease: from Immune Homeostasis to Neuroinflammatory condition. A Review

I have completed my review on manuscript ijms-1865271, entitled, Alzheimer’s Disease: from Immune Homeostasis to Neuroinflammatory condition. A Review. Although several modifiable and non-modifiable risk factors have been hypothesized, neuroinflammation has recently been considered as a major contributing component to Alzheimer's disease pathogenesis. Neuroinflammation can arise as a result of the joint activity of resident immune cells in the CNS and the adaptive peripheral immune system. Although the involvement of innate and adaptive immune cells has not been fully defined, there is evidence that immune system dysregulation plays a role in neuroinflammation. Recent studies have focused on the involvement of the adaptive immune system, demonstrating that inflammatory indicators are characterized by increased CD4+Teff cell activity and decreased circulating CD4+Treg cell activity.

The merit of this review

Overall, the study is quite informative and beneficial, indicating that immunotherapies for neurodegenerative illnesses have incorrectly concentrated on A plaques and NFT therapy in recent years. Instead, in recent years, therapeutic efforts have aimed to stimulate the spread of immunotherapies focusing on immunoregulatory, neuroprotective, neurodegenerative, and anti-inflammatory Treg functions. This technique will be a fruitful subject of research in the future, with the goal of "fighting" the progression of neurodegenerative diseases while maintaining the immune system's neuroprotective function.

The manuscript is worthy of publishing in the IJMS. I have some comments and suggestions for the authors on the current form of this manuscript that must be addressed first. The authors must pay attention to my comments and revise as suggested.

 Comments for authors regarding major revisions

Comment 1. In my opinion, the abstract is too short. Authors are recommended to expand by adding more information in the abstract.

Comment 2. In line 18 – 19, the authors write …“but several evidence suggest that immune system dysregulation play a key role in neuroinflammation.It appears that the quantifier ‘several’ does not fit with the uncountable noun ‘evidence’. Similarly ‘play’ needs to be revised as ‘plays.’

Comment 3. Use of commas after “In this review,” in line 22.

Comment 4. In line 23, “between this two systems,” need to be revised as “between these two systems.”

Comment 5. Line 36 uses the acronym "CNS" without providing the complete form. When using the abbreviated form, I recommend revising as "Central Nervous System (CNS)" the first time.

Comment 6. AD can be caused by a variety of factors. When comparing the number of factors that cause AD, the authors' description offered in the background (introduction) is insufficient to convey the information, which should be increased for new readers. I recommend that authors provide a wide spectrum of causes of AD. I have some suggestions for authors to consider in order to follow the studies and include them in the introductory portion of this review.

-          Zhang, X., Huang, W., Chen, W."Microwaves and Alzheimer's disease (Review)". Experimental and Therapeutic Medicine 12, no. 4 (2016): 1969-1972. https://doi.org/10.3892/etm.2016.3567

-          Mumtaz, S., Rana, J. N., Choi, E. H. & Han, I. Microwave Radiation and the Brain: Mechanisms, Current Status, and Future Prospects. International Journal of Molecular Sciences vol. 23 (2022). [https://doi.org/10.3390/ijms23169288].

-          Breijyeh Z, Karaman R. Comprehensive Review on Alzheimer’s Disease: Causes and Treatment. Mol. . 2020; 25. DOI:10.3390/molecules25245789.

-          Vossel KA, Tartaglia MC, Nygaard HB, Zeman AZ, Miller BL. Epileptic activity in Alzheimer’s disease: causes and clinical relevance. Lancet Neurol 2017; 16: 311–22.

Comment 6. In line 54, the word, “that is over-produced from Amyloid Precursor Protein (APP)” Is recommended to revise as “which is over-produced from Amyloid Precursor Protein (APP).”

 Comment 7. In line 60, the word, associated to neuronal is recommended to revise associated with neuronal.”

Comment 8. In line 58, the “behavioural” need to be correct.

Comment 9. In line 64, “One of the main theory” is recommended to revise One of the main theories.”

Comment 10. In line 90, the “differeces” need to be correct to “differences.”

Comment 11. In line 94, the “constitued” need to be correct. Line 97, change “appear” to “appears.”

Comment 12. In lines 126, 131, and 134, I recommend deleting the word “which.”

Comment 13. In lines 111, 169, 174, 190, 194, 206, and 252, “is able to” recommended to be replaced with “can”.

Comment 14. In line 215, “As metioned aboove” is recommended to revise As mentioned above.”

Comment 15. There is no need to repeat the full form if the short form is already defined. Replace the “Alzheimer disease” in line 388 with “AD”.

Comment 16. In line 417, the “modelwith” need to correct.

 Comment 17. In line 427, “producing” is recommended to revise production.”

Comment 18. In line 438, correct the word “summatized” with “summarized”.

Comment 19. In line 471, the “andthus” need to correct.

Comment 20: There are typos and inaccuracies in the paper. I strongly recommend authors to read precisely and correct the grammatical errors.

Author Response

REVIEWER 2

Review on Alzheimer’s Disease: from Immune Homeostasis to Neuroinflammatory condition. A Review

I have completed my review on manuscript ijms-1865271, entitled, “Alzheimer’s Disease: from Immune Homeostasis to Neuroinflammatory condition. A Review.” Although several modifiable and non-modifiable risk factors have been hypothesized, neuroinflammation has recently been considered as a major contributing component to Alzheimer's disease pathogenesis. Neuroinflammation can arise as a result of the joint activity of resident immune cells in the CNS and the adaptive peripheral immune system. Although the involvement of innate and adaptive immune cells has not been fully defined, there is evidence that immune system dysregulation plays a role in neuroinflammation. Recent studies have focused on the involvement of the adaptive immune system, demonstrating that inflammatory indicators are characterized by increased CD4+Teff cell activity and decreased circulating CD4+Treg cell activity.

The merit of this review

Overall, the study is quite informative and beneficial, indicating that immunotherapies for neurodegenerative illnesses have incorrectly concentrated on A plaques and NFT therapy in recent years. Instead, in recent years, therapeutic efforts have aimed to stimulate the spread of immunotherapies focusing on immunoregulatory, neuroprotective, neurodegenerative, and anti-inflammatory Treg functions. This technique will be a fruitful subject of research in the future, with the goal of "fighting" the progression of neurodegenerative diseases while maintaining the immune system's neuroprotective function.

The manuscript is worthy of publishing in the IJMS. I have some comments and suggestions for the authors on the current form of this manuscript that must be addressed first. The authors must pay attention to my comments and revise as suggested.

 Comments for authors regarding major revisions

Comment 1. In my opinion, the abstract is too short. Authors are recommended to expand by adding more information in the abstract. The abstract was expanded

Comment 2. In line 18 – 19, the authors write …“but several evidence suggest that immune system dysregulation play a key role in neuroinflammation.” It appears that the quantifier ‘several’ does not fit with the uncountable noun ‘evidence’. Similarly ‘play’ needs to be revised as ‘plays. Errors were corrected

Comment 3. Use of commas after “In this review,” in line 22. Corrected

Comment 4. In line 23, “between this two systems,” need to be revised as “between these two systems.” Error was corrected

Comment 5. Line 36 uses the acronym "CNS" without providing the complete form. When using the abbreviated form, I recommend revising as "Central Nervous System (CNS)" the first time.  Corrected

Comment 6. AD can be caused by a variety of factors. When comparing the number of factors that cause AD, the authors' description offered in the background (introduction) is insufficient to convey the information, which should be increased for new readers. I recommend that authors provide a wide spectrum of causes of AD. I have some suggestions for authors to consider in order to follow the studies and include them in the introductory portion of this review. The references below were introduced and additional paragraphs related to bibliographic references have been included in the text.

-          Zhang, X., Huang, W., Chen, W."Microwaves and Alzheimer's disease (Review)". Experimental and Therapeutic Medicine 12, no. 4 (2016): 1969-1972. https://doi.org/10.3892/etm.2016.3567

-          Mumtaz, S., Rana, J. N., Choi, E. H. & Han, I. Microwave Radiation and the Brain: Mechanisms, Current Status, and Future Prospects. International Journal of Molecular Sciences vol. 23 (2022). [https://doi.org/10.3390/ijms23169288].

-          Breijyeh Z, Karaman R. Comprehensive Review on Alzheimer’s Disease: Causes and Treatment. Mol. . 2020; 25. DOI:10.3390/molecules25245789.

-          Vossel KA, Tartaglia MC, Nygaard HB, Zeman AZ, Miller BL. Epileptic activity in Alzheimer’s disease: causes and clinical relevance. Lancet Neurol 2017; 16: 311–22.

Comment 6. In line 54, the word, “that is over-produced from Amyloid Precursor Protein (APP)” Is recommended to revise as “which is over-produced from Amyloid Precursor Protein (APP).” Error was corrected

 Comment 7. In line 60, the word, “associated to neuronal” is recommended to revise “associated with neuronal.”  Error was corrected

Comment 8. In line 58, the “behavioural” need to be correct Error was corrected

Comment 9. In line 64, “One of the main theory” is recommended to revise “One of the main theories.” Error was corrected

Comment 10. In line 90, the “differeces” need to be correct to “differences.” Error was corrected

Comment 11. In line 94, the “constitued” need to be correct. Line 97, change “appear” to “appears.” Error was corrected

Comment 12. In lines 126, 131, and 134, I recommend deleting the word “which.” Error was corrected

Comment 13. In lines 111, 169, 174, 190, 194, 206, and 252, “is able to” recommended to be replaced with “can”. Error was corrected

Comment 14. In line 215, “As metioned aboove” is recommended to revise “As mentioned above.” Error was corrected

Comment 15. There is no need to repeat the full form if the short form is already defined. Replace the “Alzheimer disease” in line 388 with “AD”. Corrected

Comment 16. In line 417, the “modelwith” need to correct.  Error was corrected

 Comment 17. In line 427, “producing” is recommended to revise “production.” Error was corrected

Comment 18. In line 438, correct the word “summatized” with “summarized”. Error was corrected

Comment 19. In line 471, the “andthus” need to correct. Error was corrected

Comment 20: There are typos and inaccuracies in the paper. I strongly recommend authors to read precisely and correct the grammatical errors. All the work was carefully revised

Round 2

Reviewer 3 Report (New Reviewer)

I have received a revised version of the manuscript with the author's response to my comments. The authors made some changes and improved some grammatical errors.

Comments which need to be addressed properly.

(a): The full stop needs to be removed from the title.

(b): In response to my previous comment 6, the authors state that they have made changes to the manuscript and have included the suggested references. Unfortunately, the author's response letter does not assist in locating the changes in the text, and the references are also missing. I recommend that authors include the page, line number, and reference number where they made changes based on the comments. To strengthen the background information for new readers, the suggested references should be included in this review.

(c): I suggest that the authors resend the response letter after making improvements and indicating the location of changes in the response letter so that I can easily track the changes.

Author Response

Enclosed please find the revised version of the manuscript , entitled Alzheimer’s Disease: from Immune Homeostasis to Neuroinflammatory condition. A Review., by Lucia Princiotta Cariddi*, MD PhD, Marco Mauri, MD PhD, Marco Cosentino, MD PhD, Maurizio Versino MD, Franca Marino, BSc PhD

                        All manuscript and references were carefully revised and completely restructured, according to Referee’s suggestions. A point-by-point response to the Reviewer’ comments is enclosed hereafter, and changes included in the text are outlined in yellow throughout the manuscript, while removed sentence are in red and crossed out. A cleaned version of the revised manuscript is included.

            We would like to express our gratefulness to the Editor and the Reviewer for their comments and suggestions, which undoubtedly helped us to improve the quality of our study, and hope that the present version of the manuscript is suitable for publication.

Enclosed the point- by -point response to referee’s criticisms.

Sincerely,

Lucia Princiotta Cariddi

* Lucia Princiotta Cariddi

PhD Program in Clinical and Experimental Medicine and Medical Humanities,

University of Insubria

Via Ottorino Rossi n. 9

21100 Varese VA - Italy

Phone +39 0332 393512

Fax +39 0332 217409

E-mail: [email protected]; [email protected]

Comments which need to be addressed properly.

(a): The full stop needs to be removed from the title.

The full stop was been removed

(b): In response to my previous comment 6, the authors state that they have made changes to the manuscript and have included the suggested references. Unfortunately, the author's response letter does not assist in locating the changes in the text, and the references are also missing. I recommend that authors include the page, line number, and reference number where they made changes based on the comments. To strengthen the background information for new readers, the suggested references should be included in this review.

            We apologise with the Reviewer. Unfortunately we re-submitted a version of revision 4 that was not the one that already included the modification concerning the references

- “Zhang, X.; Huang, W.-J.; Chen, W.-W. Microwaves and Alzheimer’s Disease. Exp Ther Med 2016, 12, 1969–1972, doi:10.3892/etm.2016.3567”.  

In the current version, this reference is # 41, page 4, line 146; the corresponding  changes in the text are at the lines 143,144 and 145.

- “Mumtaz, S.; Rana, J.N.; Choi, E.H.; Han, I. Microwave Radiation and the Brain: Mechanisms, Current Status, and Future Prospects. Int J Mol Sci 2022, 23, 9288, doi:10.3390/ijms23169288”.

In the current version, this reference is # 42, page 4, line 146; the corresponding changes in the text are at the lines 143,144 and 145.

-Breijyeh, Z.; Karaman, R. Comprehensive Review on Alzheimer’s Disease: Causes and Treatment. Molecules 2020, 25, 5789, doi:10.3390/molecules25245789.

In the current version, this reference is # 43, page 4, line 146; the corresponding changes in the text are at the lines 143,144 and 145.

-Vossel, K.A.; Tartaglia, M.C.; Nygaard, H.B.; Zeman, A.Z.; Miller, B.L. Epileptic Activity in Alzheimer’s Disease: Causes and Clinical Relevance. Lancet Neurol 2017, 16, 311–322, doi:10.1016/S1474-4422(17)30044-3.  

In the current version, this reference is # 40, page 4, line 143; the corresponding  changes in the text are at the lines 138, 139, 140 141, 142, 143

c): I suggest that the authors resend the response letter after making improvements and indicating the location of changes in the response letter so that I can easily track the changes.

            We hope that the current versions of both our text and our response letter full fill the Reviewer’s requirements.

Round 3

Reviewer 3 Report (New Reviewer)

In this version, the authors addressed all of my comments and concerns. In its present form, the paper deserves to be published in IJMS.

This manuscript is a resubmission of an earlier submission. The following is a list of the peer review reports and author responses from that submission.

Round 1

Reviewer 1 Report

This review focuses on innate and adaptive immune cells and their neuroinflammatory responses in Alzheimer’s disease pathogenesis.

Abstract needs rephrasing and English check for grammar –

i.e “Many modifiable and non-modifiable risk factors…”

“with focus on how the cross-talk between two systems….”

Line 30-41: Multiple paragraphs in the introduction is not necessary. Is this a formatting error? Please check.

Line 30: First line is not necessary, unless it can be combined with the second line 32.

Introduction is hard to read should be expanded and better linked to subsequent sentences.

Line 50-64: Multiple paragraphs is not necessary. Is this a formatting error? Please check.

Paragraph 2.1: An illustration depicting the amyloid cascade hypothesis would be great since little is explained about the process.

Line 90 : “In AD Tau proteins are hyperphosphorylated and abnormally aggregated, and lose their ….”– please check grammar

Line 103: “Microglial cells delete Ab peptide…” this is perhaps not the best way to describe a scientific process. Can phagocytosing be considered?

Line 108-109: “were multiple and different…” do you mean multifactorial?

Line 101: “representing the main reason” .. this sentence needs rephrasing

Line 159: “amyloid-beta plaques” can be abbreviated to Aβ

Line 129: “however these treatments were found not to be curative, nor they were modifying” do you mean - nor were they modifying?

Line 178: “are the major represented class of macrophages..” This line needs rephrasing.

The title of 3.1 “The Physiological Role of CNS Innate Immune Cells” should be similar to 3.2 “Physiological Role of Peripheral Adaptive Immune Cells in CNS”. Please choose a consistent format.

Section 3.1: Although astrocytes aid in regulating the innate immune response, it is considered as glial cells, not innate cells. Paragraph about astrocyte should be removed. Alternatively, change the title to reflect the inclusion of glial cells.

Line 172: Not quite sure where did the authors mention that the innate immune system consist of astrocytes before this paragraph?

Line 218: “Usefull” authors meant useful?

Line 219: “usefull to catch circulating antigens”. This not the best way to describe a scientific process, please rephrase catch.

Line 220: “tending to be under” authors meant tend?

Section 3.2: Topic on B cells needs to be expanded.

Line 302: “wich”

Line 302: Authors need to rephrase this sentence, as it does not make sense.

Section 4.2: Innate cells like microglia play a massive role in AD pathogenesis. However, Authors need to expand on microglial association with Amyloid plaques, and expand on studies/findings from high throughput studies and animal models.

Line 307: “disease-dipendent”. Authors meant dependent?

Section 4.1: Microglia and aging implicates AD. This section needs to be expanded by describing findings from AD models.  

Line 332: Authors, please check English.

Table 1 references are not numbered – Authors please check.

Authors mention about clinical evidences in the conclusion but did not expand on clinical findings throughout the review.

Overall, this review is limited in identifying the current gap in the knowledge and does little to provide additional information to the field. Multiple English errors and formatting have thus made the review very difficult to read and comprehend key ideas of paragraphs.

Author Response

1 - REVIEWER

(Comments and Suggestions for Authors

This review focuses on innate and adaptive immune cells and their neuroinflammatory responses in Alzheimer’s disease pathogenesis.

Abstract needs rephrasing and English check for grammar –

i.e “Many modifiable and non-modifiable risk factors…”

“with focus on how the cross-talk between two systems….”

Abstract was revised

Line 30-41: Multiple paragraphs in the introduction is not necessary. Is this a formatting error? Please check.

Line 50-64: Multiple paragraphs is not necessary. Is this a formatting error? Please check.

Multiple paragraphs were removed through the whole text

Line 30: First line is not necessary, unless it can be combined with the second line 32.

Removed and sentence changed

Introduction is hard to read should be expanded and better linked to subsequent sentences.

Introduction was revised and expanded in some parts

Paragraph 2.1: An illustration depicting the amyloid cascade hypothesis would be great since little is explained about the process.

Figure was added

Line 90 : “In AD Tau proteins are hyperphosphorylated and abnormally aggregated, and lose their ….”– please check grammar

revised

Line 103: “Microglial cells delete Ab peptide…” this is perhaps not the best way to describe a scientific process. Can phagocytosing be considered?

Sentence was changed and appropriate term introduced

Line 108-109: “were multiple and different…” do you mean multifactorial?

Sentence changed

Line 101: “representing the main reason” .. this sentence needs rephrasing

rephrased

Line 159: “amyloid-beta plaques” can be abbreviated to Aβ

Abbreviation was introduced

Line 129: “however these treatments were found not to be curative, nor they were modifying” do you mean - nor were they modifying?

Error corrected

Line 178: “are the major represented class of macrophages..” This line needs rephrasing.

rephrased

The title of 3.1 “The Physiological Role of CNS Innate Immune Cells” should be similar to 3.2 “Physiological Role of Peripheral Adaptive Immune Cells in CNS”. Please choose a consistent format.

Titles were changed

Section 3.1: Although astrocytes aid in regulating the innate immune response, it is considered as glial cells, not innate cells. Paragraph about astrocyte should be removed. Alternatively, change the title to reflect the inclusion of glial cells.

This section was revised

Line 172: Not quite sure where did the authors mention that the innate immune system consist of astrocytes before this paragraph?

The sentence was revised

Line 218: “Usefull” authors meant useful?

Error corrected

Line 219: “usefull to catch circulating antigens”. This not the best way to describe a scientific process, please rephrase catch.

 rephrased

Line 220: “tending to be under” authors meant tend?

 Error corrected

Section 3.2: Topic on B cells needs to be expanded.

 Section was expanded

Line 302: “wich”

Line 302: Authors need to rephrase this sentence, as it does not make sense.

Sentence was rephrased

Section 4.2: Innate cells like microglia play a massive role in AD pathogenesis. However, Authors need to expand on microglial association with Amyloid plaques, and expand on studies/findings from high throughput studies and animal models.

This section was revised and expanded

Line 307: “disease-dipendent”. Authors meant dependent?

 Error corrected

Section 4.1: Microglia and aging implicates AD. This section needs to be expanded by describing findings from AD models.  

This section was revised and expanded

Line 332: Authors, please check English.

English was revised through the whole text

Table 1 references are not numbered – Authors please check.

Table was revised

Authors mention about clinical evidences in the conclusion but did not expand on clinical findings throughout the review.

This section was revised and expanded

Overall, this review is limited in identifying the current gap in the knowledge and does little to provide additional information to the field. Multiple English errors and formatting have thus made the review very difficult to read and comprehend key ideas of paragraphs.

Reviewer 2 Report

The study by Lucia et al. reviewed role of both innate and adaptive immune systems in neuroinflammatory pathways of Alzheimer’s Disease progression. This is interesting manuscript however, there are few minor corrections to be made before the final acceptance of the manuscript.

  1. In Fig 1 a,b depict in a way to show the change between healthy and damaged neuron. Increase the resolution of the figures so that the cytokine names are visible. Also give the figure shape legend details for microglia types and other cell types illustrated.
  2. Suggestion: Add few more literature or make subsection for BBB damage/dysfunction in AD.
  3. In line 76, change respectly to respectively
  4. In line 106, citation is required to support neuroinflammatory state
  5. Too many double spacing in the Manuscript – modify accordingly
  6. In the subsection 3.2, the title topic is about CNS “3.2. Physiological Role of Peripheral Adaptive Immune Cells in CNS” but in fact there is nothing relevant to CNS in the description – modify accordingly
  7. In line 223, T-cell receptor (TCR) not Toll- like receptors
  8. In line 249, conversely not conversly
  9. In line 269, CD4+ T cells not CD4 T cells
  10. In line 301, the cited reference number 97 is not relevant to the text– modify accordingly. Double check the references cited to the relevant texts throughout the manuscript.
  11. In line 307, dependent not dipendent
  12. In line 307, check for the Grammar
  13. In the subsection 4.2, Aβ peptide not Aβ protein– modify accordingly
  14. In lines 360,369,391,398,247- consider improving scientific English writing.
  15. Be consistent with reference format used through out the manuscript. In line 465 it is not numbered. Also in the table 1 be consistent with reference format.
  16. In line 513, to instead of to to .

Round 2

Reviewer 1 Report

Dear authors, major changes to English is necessary to convey meanings of the review and studies. Please consider having this manuscript rewritten.

Change “not-modifiable risk” to non-modifiable risk

Line 19: “has” to have

Figure 1: “Hypotesis” is spelt wrong

Figure 1 title Amyloid cascade hypothesis does not represent the illustration and is too simple to describe the process of the amyloid cascade. Either change the title to how amyloid beta is formed in the brain or better explain what the amyloid cascade is – include pathways. Also, Ab1-42 leads to the formation of Ab oligomers and Ab monomers.

Line 118: “cuase

Line 20: “represnert

Line 101: “and begin”?

Section 2.2- Since NFT is abbreviated, please be consistent throughout the section/manuscript.

Line 101: Authors mentioned three stages is required for neurofibrillary tangles. Please elaborate those stages.

Line 109: fibrils tangles is referring to NFT, please be consistent.

Line 118: please replace don’t with do not. Replace “cuase” to cause

Line 120: “represnert”

Line 128-136: Can authors please elaborate on these theories.

Line 137: “plausibel”

Line 137-143: Can authors please describe the therapeutic studies done for AD. What is the treatment that failed?

Line 143: Elaborate what treatment?

Line 147-149: What other pathological mechanisms are you referring to? Please elaborate.

Line 149- “As mentioned before”. Authors did not mention the mechanism prior to this sentence, please rephrase.

Line 151: “showed”

Line 161: “found”. Please rephrase/clarify 160-163. This sentence is confusing.

Line 165: “this latter” needs to be deleted.

Line 166: CSF has not been abbreviated before.

Line 172: “amyloid-beta” stick with abbreviated form

Line 173: “delete waste” is not a suitable word

Line 181: “of, of”

Line 186: “immuen”

Line 187: “during adult age”. This sentence is misplaced.

Line 195: “capable of recruiting”

Line 220: “On the other hand”

Line 230: “they move” – what cells are you referring to?

Line 233: “in these areas”. Please specify what areas

Line 234: authors please describe what direct and indirect pathway is in detail

Line 240: “TLR” has not been previously abbreviated

Line 244: please describe in detail direct and indirect method of T cell “killing” infected cells.

Line 244, section, should be described before line 238

Line 250 need references to substantiate claims

Line 251-259,260-274 need references to substantiate claims. Additional studies need to be included as support.

Line 267: “citokynes”

Line 268: Th17 indeed play an important role. Authors need to discuss this further, include studies.

Line 270: These conditions need to be expanded for the benefit of the readers/review.

Section 3.2 gives the reader a very brief summary of adaptive immune cells role in inflammation. This is insufficient as a review, please provide studies to support/refute claims.

Line 287: “It is”

Lin 292: Authors mentioned several studies but cited a reference. More references is required to substantiate claims.

Section 4.1 needs to be expanded.

Line 292 and Line 340: This is a copy and paste sentence. Authors please check! This is a major issue.

Line 344-348 needs to be rephrased.

Line 355: it is not clear how microglia cell death can increase Ab plaque formation. Please elaborate.

Line 378 : “as mentioned before” and Line 449 “As said before” is not necessary. Please remove throughout the manuscript. Line 378-384 repetitive as this is similar to line 371.

Line 383 is better explained after lines 271-274

Line 383: “APC” has not been previously abbreviated

Line 389: elaborate on “injurious conditions” and “move-on in”

Line 472: Adaptive immunity in AD is well-described: PMC4780638  

Line 472 -474: Sentence needs rephrasing-check English “In comparison to Parkinson's Disease, however, in AD peripheral adaptive immunity has received less attention and less is currently known about the pattern profile of immune dysregulation, that occurs in AD.

Line 545: Parkinson’s disease has been abbreviated.

Table 1: “in” animal and human models.

Table 1 ref is not formatted to number, column not aligned

Line 538-540: sentence needs rephrasing

Line  546: “strokes” you mean stroke

Line 549-552: needs rephrasing

Line 552: “These last” not too sure what this sentence mean

Line 554: Please provide studies and expand on how T cells regulate brain homeostasis.

Line 560: “however” is not necessary

Line 577: Authors please elaborate on what “clinical evidences” refers to?

Line 580: authors please elaborate on “focused wrongly on targeting protein aggregates”

Round 3

Reviewer 1 Report

Dear authors

Major revision of the manuscript is unfortunately still required.

There are (still) various grammar and sentence issues throughout the manuscript (i.e. Line124: Alluminium, line 209, Line 217, line 243, line 245, line 278, line 279, line 348-353 and more). Abbreviations were not checked throughout the manuscript (i.e b-amyloid)

Repetitive statements are also noted throughout the manuscript, (i.e Lines 217-224 and lines 323-328).

Statements throughout the manuscript should be elaborated (i.e 344, "some changes") - Gene alterations associated with mentioned process needs to be elaborated to give the reader a better understanding of the genes involved in the process.